# Acellular Biomaterials Associated with Autologous Bone Marrow-Derived Mononuclear Stem Cells Improve Wound Healing through Paracrine Effects

**DOI:** 10.3390/biomedicines11041003

**Published:** 2023-03-24

**Authors:** Isio Carvalho de Souza, Aline Luri Takejima, Rossana Baggio Simeoni, Luize Kremer Gamba, Victoria Stadler Tasca Ribeiro, Katia Martins Foltz, Lucia de Noronha, Meila Bastos de Almeida, Jose Rocha Faria Neto, Katherine Athayde Teixeira de Carvalho, Paulo Cesar Lock da Silveira, Ricardo Aurino Pinho, Julio Cesar Francisco, Luiz César Guarita-Souza

**Affiliations:** 1Experimental Laboratory of Institute of Biological and Health Sciences, Pontifícia Universidade Católica do Paraná (PUCPR), 1555 Imaculada Conceição Street, Curitiba 80215-901, SP, Brazil; 2Department of Veterinary Medicine, Universidade Federal do Paraná (UFPR), Rua XV de Novembro, 1299, Curitiba 80060-000, SP, Brazil; 3Cell Therapy and Biotechnology in Regenerative Medicine Department, The Pelé Pequeno Príncipe Institute, Child and Adolescent Health Research & Pequeno Príncipe Faculties, 1632 Silva Jardim Avenue, Curitiba 80240-902, SP, Brazil; 4Laboratory of Experimental Pathophysiology, Graduate Program in Health Sciences, Universidade do Extremo Sul Catarinense, Criciúma 88806-000, SC, Brazil; 5Laboratory of Exercise Biochemistry in Health, School of Medicine, Graduate Program in Health Sciences, Pontifícia Universidade Católica do Paraná (PUCPR), 1555 Imaculada Conceição Street, Curitiba 80215-901, SP, Brazil

**Keywords:** human acellular amniotic membrane, experimental animal models, antioxidant system, stem cells, wound healing, paracrine

## Abstract

Wound healing is a complex process of repair that involves the interaction between different cell types and involves coordinated interactions between intracellular and extracellular signaling. Bone Marrow Mesenchymal Stem Cells (BMSCs) based and acellular amniotic membrane (AM) therapeutic strategies with the potential for treatment and regeneration of tissue. We aimed to evaluate the involvement of paracrine effects in tissue repair after the flap skin lesion rat model. In the full-thickness flap skin experiment of forty Wistar rats: A total of 40 male Wistar rats were randomized into four groups: group I: control (C; n = 10), with full-thickness lesions on the back, without (BMSCs) or AM (n = 10); group II: injected (BMSCs; n = 10); group III: covered by AM; group IV–injected (AM + BMSCs; n = 10). Cytokine levels, IL-1, and IL-10 assay kits, superoxide dismutase (SOD), glutathione reductase (GRs) and carbonyl activity levels were measured by ELISA 28th day, and TGF-β was evaluated by immunohistochemical, the expression collagen expression was evaluated by Picrosirius staining. Our results showed that the IL-1 interleukin was higher in the control group, and the IL-10 presented a higher mean when compared to the control group. The groups with BMSCs and AM showed the lowest expression levels of TGF-β. SOD, GRs, and carbonyl activity analysis showed a predominance in groups that received treatment from 80%. The collagen fiber type I was predominant in all groups; however, the AM + BMSCs group obtained a higher average when compared to the control group. Our findings suggest that the AM+ BMSCs promote skin wound healing, probably owing to their paracrine effect attributed to the promotion of new collagen for tissue repair.

## 1. Introduction

Wound healing is a complex process of repair that involves the interaction between different cell types and biochemical and physiological events, such as the proliferation and remodeling of tissue [1].

This process involves five main distinct stages: hemostasis, inflammation, angiogenesis, growth, proliferation, and remodeling. Wound repair also involves specialized cells in regeneration, the formation of granulation tissue (monocytes, basophils, eosinophils, and neutrophils), and the reconstruction of tissue which is triggered in inflammatory phases and supported by biochemical mediators [2].

Studies suggest that the wound healing process occurs through epidermal growth factor (EGF), insulin-like growth factor 1 (IGF-1), platelet-derived growth factors (PDGF), the transformation of growth factor β (TGF-β), and vascular endothelial growth factor (VEGF) and protein in the extracellular matrix (ECM) proteins such as collagen, fibrin, fibronectin that stimulate wound healing by indirectly enhance inflammation and tissue repair [3,4].

Although, there are many barriers to solve, such as the rapid degradation of growth factors. Numerous strategies and biomaterials have been applied to replace skin damage. Finding the ideal material for skin replacement is a field of skin tissue engineering. The ideal material for skin replacement should be its stationary biocompatibility and degradability is essential for performing and maintaining essential cellular activities, which include the ability of cell adhesion and proliferation [5]. Since 1940, the amniotic membrane has been widely used as an alternative in wound healing and various areas of tissue engineering and regenerative medicine [6,7].

The amniotic membrane has shown great potential as an ideal biomaterial due to its growth factors and biological structure, which contains appropriate microarchitecture, epithelialization stimulation capacity, antibacterial properties, and lower immunogenicity [8].

Additionally, it has been demonstrated in clinical studies that the use of an acellular amniotic membrane (AM) improves the healing process due to bioactive factors, proteases, including a large group of cytokines and effective anti-inflammatory proteins, which remain intrinsic after the decellularization process that can be used physical, chemical, and biological methods, such as VEGF, TGFb1, bFGF and EGF [9].

In recent years, many studies have shown to exhibit through their bioinductive properties, and the acellular membrane may also promote epithelialization while suppressing inflammation, angiogenesis, vasculogenesis, and scarring [10,11]. In the field of cell therapeutic strategies, stem cell-based regenerative therapies have been successfully developed that facilitate skin repair and regeneration; stem cells have been reported as a cellular source in regenerative medicine.

Similar to other stem cells, Bone Marrow Mesenchymal Stem Cells (BMSCs) can differentiate into local components of tissue injury and have paracrine factors and cytokines capable of activating tissue survival, repair, and regeneration [12]. Thus, recent studies reported that BMSCs and paracrine mechanisms together might be key to the regeneration of damaged tissue [13]. This study was conducted to evaluate the role of paracrine interaction of Bone Marrow Mesenchymal Stem Cells (BMSCs) associated with acellular amniotic membrane (AM) in the wound healing pre-clinic model.

## 2. Materials and Methods

### 2.1. Animals

All the experimental procedures were performed with the approval of the Ethics Committee (CEP) and Ethics Committee for the Use of Animals (CEUA), under approval number 01238 of the Experimental Medical school of Pontifícia Universidade Católica do Paraná, and all procedures conformed to the NIH guidelines. The animals were housed in individual cages and maintained in an environmentally controlled laboratory (12-h light/dark cycle at 25 °C), with access to food and water *ad libitum* [7].

### 2.2. Experimental Design

Forty male *Wistar* rats weighing between (250–350 g) were used in this study. The animals were randomized into four groups: group I: control (C; n = 10), with full-thickness lesions on the back, without (BMSCs) or AM (n = 10); group II: injected (BMSCs; n = 10); group III: covered by AM; group IV: injected (AM + BMSCs; n = 10). The animals were subjected to random skin flaps, and the BMSCs and AM were transplanted. Thirty days after surgery, all animals were analyzed by the viability of flaps that were determined by the clinic (color and capillaries) and histological analysis. All rats were euthanized by a lethal dose of thiopental sodium (100–250 mg/kg, i.p.), and histopathological examinations were performed (Figure 1).

### 2.3. Isolation and Characterization of BMSCs/Acellular Amniotic Membrane Preparation

Human AM was acquired from aseptic sites from 1 woman who received elective cesarean sections at Maternidade Victor Ferreira do Amaral Maternal and Child Health Hospital (Curitiba, Brazil). The patients were informed a priori and consented to donation. The study was conducted conformed to tenets of the declaration of Helsinki. The BMSCs were obtained by the iliac crest of male Wistar rats (250–300 g) as previously described [10]. Centrifugation was applied to isolate BMSCs after aspirating into the iliac crest of each rat. Briefly, the BMSCs were collected and centrifuged at 1400× *g* rpm for 40 min and diluted in essential Dulbecco’s modified Eagle medium (DMEM), gently separated by the Ficoll-Hypaque density gradient method (density = 1.077 g/mL). Subsequently, sediment after 2 washing steps, at 1500× *g* rpm for 10 min, obtained was resuspended with DMEM and the cell count was performed in the Neubauer chamber. The validation of cell viability was verified utilizing Trypan blue stain.

Decellularization was performed using an aseptic technique in a Class II BioSAFE (Veco^®^) biological. Thereafter, it was treated with process chemical and mechanical using 0.01% SDS solution (sodium dodecyl sulfate) and SD (sodium deoxycholate) at 0.01% for 24 h at 37 °C in pH 7.2 saline phosphate buffer (PBS), with the aid of a stirrer mechanic (Shaking Table 109 M, Nova ÉticaLtda, Brazil). Then preserved in PBS (Gibco) at 4 °C, according to the methodology described by Blume et al., 2021 [7].

### 2.4. Flow Cytometry Cell Analysis

Flow cytometry was performed to determine the expression of BMSC cellular face markers was performed on freshly isolated mononuclear cells and on adherent cell populations. Cells were trypsinized and resuspended in PBS for 30 min at 4 °C were incubated for 2 h at room temperature in the dark with the following immunophenotypic analyses for CD34, CD 45, CD105, CD 90, CD73, and CD105 were performed with a commercially available kit (Stem Kit, Beckmann Coulter, Krefeld) and mouse IgGk1 isotype controls (BD Pharmingen, 5 μg/mL) were used as control primary antibodies. We use all of the analysis tools by a CyFlow™ Space flow cytometer (Sysmex Partec GmbH) with FloMax 2.8 software (Sysmex Partec GmbH).

### 2.5. Rat Flap Skin Model and Membrane Implant

The surgical flap preparation technique was applied as proposed by Takejima et al., 2021 [14]. The animals were intraperitoneally anesthetized with ketamine hydrochloride 10% at a dose of 90 mg/kg and xylazine 2% at a dose of 10 mg/kg, the animals where dorsum trichotomy was performed, and the animal was performed with an area of approximately 5 × 3 cm rectangular mold (stamp), the area where the cutaneous lesion was to be induced in the dorsal region between the 2 scapulae was delimited, with the largest axis in the longitudinal direction. The rats were then randomly divided into 4 groups of ten each, as described above. BMSCs (1 × 10^7^ total cells) or BMSCs + AM were injected in eight areas (subcutaneously administered 1 × 10^7^ cells/area for eight areas in all wound edges). The suspension of cells or vehicles was applied along the margin of the dorsal wound. The flap was covered with a Vaseline rayon dressing that was sutured with a simple 5–0 nylon stitch on the four vertices of the lesion to avoid contamination and manipulation by other animals. These dressings were removed after 2 days of evolution. All rats received 20 mg/kg ibuprofen and then repeated every 12 h for 3 days (Figure 2).

The animals were submitted to injection of BMSCs on the skin defect. The flaps were sutured with a 4.0 polypropylene thread. After the sin flap was re-suture, the skin tissues were covered.

### 2.6. Measurements of Cytokines, IL-1, IL10, and TGF-β Levels

The IL-1 assay kit IL-10 and TGF-β assay kit (cat. no. DY522-05) were purchased from Cloud-Clone Corp (Cloud-Clone Corp Technology co., Ltd., Wuhan, China). Briefly, for IL-1 and IL-10 assay, 100 µL of a standard blank or samples were added into the appropriate wells and then incubated for 1 h at 37 °C. After aspiration, 100 µL of prepared detection reagent was added to each well and incubated for 1 h at 37 °C. After aspiration and washing, 90 µL of substrate solution was added to wells and incubated for 15 min at 37 °C. The optical density (OD) of the reaction product was read on a microplate reader (Sigma-Aldrich, St. Louis, MO, USA) at 450 nm [15].

### 2.7. Determination of the Oxidative Damage

Protein carbonylation was determined according to Levine et al. [15]. Briefly, carbonyl group levels were estimated spectrophotometrically at 412 nm using 2,4-dinitrophenylhydrazine (DTNB), and the amounts of TNB formed (equivalent to the amounts of carbonyl were calculated).

### 2.8. Superoxide Dismutase Activity and Glutathione Reductase

To measure the superoxide dismutase (SOD) and glutathione reductase (GRs) levels, the dorsum partial skin tissues were immediately put into ice-cold normal saline containing 50 U/mL aprotinin. The tissue homogenate (10%, *w*/*v*) was prepared and centrifuged at 1200× *g* for 10 min. The supernatant was measured by the ELISA method following the manufacturer’s protocols, using the absorbance at 480 nm [16].

### 2.9. Measurements of Protein

The protein content from skin tissue homogenates was assayed using bovine serum albumin as a standard according to the method described by Lowry et al. [17]. Folin phenol reagent (phosphomolybdic-phosphotungstic reagent) was added to the protein homogenate. The bound reagent was slowly reduced, changing from yellow to blue. The absorbance was read at 480 nm.

### 2.10. Euthanasia

All Animals were euthanized with a lethal dose of pentobarbital Sodium Thiopental (100 to 250 mg/kg) injected intraperitoneally. Then, their skin tissues were carefully removed and frozen at −80 °C for the determination of cytokine protein and oxidative stress.

### 2.11. Immunohistochemistry Analysis

Immunohistochemistry was performed using the avidin-biotin complex method https://www.abcam.com/abpromise (accessed on 28 February 2023) as described in our previous study [16]. Rabbit polyclonal to TGF-β Receptor I was used as primary antibodies. Sections were then incubated with the appropriate biotinylated secondary antibody. Peroxidase activity was determined using a diaminobenzidine (DAB) substrate (Abcam, Cambridge, MA, USA).

### 2.12. Histopathological Analysis

The sections were stained with hematoxylin-eosin (HE) and Picrosirius Red for histological evaluation and fixed in 10% paraformaldehyde, and embedded in paraffin. Then, 5-μm-thick sections were cut from the blocks, affixed to glass slides coated with poly-L-lysine, examined under a microscope (Nikon Eclipse E400; Nikon, Tokyo, Japan), and evaluated with a digital image analysis system. Immunohistochemically analysis was performed as previously described by Jones et al. 2008 [18]. Briefly, the skin samples were incubated with a primary antibody applied for 1 h at room temperature. Tissue sections were then incubated with TGF-β primary antibody (1:200, Abcam Inc; http://www.abcam.com, accessed on 28 February 2023) for 1 h at 25 °C and incubated with primary antibodies overnight at 4 °C, followed by incubation with secondary antibodies for 2 h at 25 °C, followed by washing and visualized using an isothiocyanate-conjugated secondary antibody (Jackson Immuno Research Laboratories Inc., West Grove, PA, USA) and imaged with an optical microscope Axio Scan.Z1 scanner (Carl Zeiss, Germany).

### 2.13. Statistical Analysis

The results of the markers were described by mean ± standard deviation (SD) and amplitude interquartile. For the comparison of the groups, in relation to variables that met the normality condition, the variance analysis model (ANOVA) was used with a factor and the Bonferroni posthoc test. The other variables were analyzed using the nonparametric Kruskal–Wallis test and Dunn’s posthoc test corrected by Bonferroni. The condition normality of the variables was evaluated by the Shapiro-Wilk test. Statistical significance was defined as *p* <  0.05. All analyses were accomplished using the software in SPSS statistical software version 25 (SPSS Inc., IBM, Chicago, IL, USA).

## 3. Results

Forty animals were initially included in the study; one died in group II on the 7th postoperative day during anesthetic induction. Thus, 39 animals were included and randomized into four groups: four groups: group I: control © (n = 10), with full-thickness lesions on the back, without (BMSCs) or AM (n = 10); group II: injected (BMSCs; n = 9); group III: covered by AM; group IV: injected (BMSCs) and covered by AM (n = 10).

### 3.1. Characterization of BMSCs

BMSCs expression of the markers CD105, CD 90, CD73 and CD105, respectively; and for hematopoietic markers CD45 (Pro-B) and CD34(Lymphocyte) showed negativity, respectively, which indicates the purity of MSC, using a previously published protocol [14] Figure 3.

### 3.2. Assessment of H&E Staining Findings

Evaluation of H&E staining to detect the presence of re-epithelialization, angiogenesis, and fibrosis with remodeling, evidenced by a well-formed epidermis Figure 4.

### 3.3. Comparison of Interleukin Protein 1 (IL1) Concentration in Tissues Adjacent to the Skin Wound in Rats

To investigate the comparison of interleukin protein 1 (IL1) concentration in tissues adjacent to the skin wound compared to that in the normal controls. IL-1 expression revealed a lower expression in groups BMSCs and BMSCs + AM (3.615 ± 1.28 and 2.817 ± 0.82; *p* = 0.960). No significant variation in IL-1 was observed in the BMSCs and AM group only (Figure 5).

### 3.4. Comparison of Interleukin Protein 10 (IL10) Concentration in Tissues Adjacent to the Skin Wound in Rats

The results of these experiments are shown in (Figure 6). In the intergroup analysis after 28 days, the IL-10 expression revealed a lower expression in groups BMSCs and BMSCs + AM (13.961 ± 10.06 and 15.479 ± 3.81; *p* = 0.765). No significant variation in IL-10 was observed in the BMSCs and AM group only.

### 3.5. Comparison of Interleukin Protein (TGF-β) Concentration in Tissues Adjacent to the Skin Wound in Rats

The immunostaining analysis performed in the present study is represented in Figure 4. The expression of TGF-β in skin tissue tended to decrease on day 28, compared with the control group, but this was not significant (Figure 7).

Results are shown as mean ± standard deviation [interquartile range]. *p* < 0.05 denoted no statistical significance in comparison to the control group or intragroup 28th-day analysis (*). TGF-β: *Transforming growth factor-beta (TGF-β)*; BMSCs: bone marrow mononuclear stem cells; *AM* (acellular amniotic membrane).

### 3.6. Oxidative Damage Markers

We evaluated the effect of BMSCs + AM on flap skin protein carbonylation content. BMSCs + AM decreased skin protein carbonylation by 80% (*p* < 0.001) and moderately decreased BMSCs and AM group at 28 days as compared to the control group. (0.156 ± 0.026 nmol/mg protein; Figure 8).

### 3.7. Antioxidant System

Twenty-eight days post-surgery, skin tissue SOD Activity samples were found to be significantly lower in the control group compared to BMSCs + AM (*p* < 0.001). Significant differences were observed between the levels of SOD in the skin tissues of C compared to the other groups (*p* > 0.05), as shown in (Figure 9).

### 3.8. Collagen Level Analysis

After 28 days of treatment, the concentration of type I collagen is a significant increase in all four groups, and the relative expression of collagen type III is the highest average (81.5 ± 3.8, *p* = 0.57) in the control group when compared to other groups. This indicates a capacity for tissue regeneration, suggested by the development of immature collagen and the formation of tissue (Figure 10).

## 4. Discussion

We used a model of flap wounds in rats to evaluate the role of paracrine interaction of Bone Marrow Mesenchymal Stem Cells (BMSCs) in the wound healing and oxidative stress parameters 28th postoperative. Histological analysis showed that BM-MSC treatment re-established the epidermal epithelization and atrophy induced by surgery.

BM-MSCs possibly acted on epithelialization, increasing the proliferation of epidermal cells residing in the participation of epidermal growth factor, probably developed by bone marrow cells or by their differentiation in epidermal cells. Our findings indicate that the acellular matrix combined with BMSCs cells results in a significant increase in the initial postoperative formation of collagen fibers and elastic fibers after skin reconstruction, probably due to their anti-inflammatory effect on cells or fusion between them.

Wound healing is a complex system that provides the molecular mechanisms of cell-cell interactions, cell-matrix interactions, cellular proliferation, matrix formation, remodeling, and tissue organization [19]. Clinical trials and preclinical studies have used amniotic membrane (AM) in the treatment of various diseases such as chronic skin wounds and thermal burns because AM contains growth factors, proteases, and fibrosis inhibitors, conferring its anti-inflammatory and antimicrobial effects, which are crucial for regenerative medicine [20,21].

IL-1β is a pro-inflammatory cytokine that plays an important role in the early phase of inflammation in the wound-healing process, stimulating the production of extracellular matrix proteins [22]. As well, IL-10 plays a vital role in cell healing, wound healing, tissue repair, angiogenesis, and antifibrotic tissue repair properties being a vital reparative cytokine. IL 10 is an anti-inflammatory that is involved in the wound healing process and inhibition of the synthesis of proinflammatory cytokines such as IL-1β, among other macrophages activated in the inflammatory process. In the present study, the IL-1β expression after treatment for 28 days suggested decreased inflammation, although the level the IL-10 showed no significant differences between the treated group and the control group, which facilitated healing [23].

Some studies have revealed that AM and BMSCs can express angiogenic factors, which is crucial to tissue flap regeneration and a complex process; it involves the proliferation of endothelial cells and the cooperation of various growth factors [24,25,26]. In the current study, immunohistochemical staining together with the cytokine level showed that the IL-1 and IL-10 were overexpressed at the late stages of healing (28 days postoperative); these data corroborate with the literature.

The possible mechanism of the therapeutic effects of AM + BMSCs on skin fibrosis might be attributed to the stimulation of TGF-β which is considered a favorable regulator of angiogenesis via the recruitment of inflammatory cells, such as macrophages, monocytes, and neutrophils, due to the up-regulation of various chemokines expressions which modulates the formation of fibroblasts, which in turn are responsible for the healing process [27]. Similar to the developmental roles of interleukins, TGF-β is implicated in stimulating the paracrine mechanism BMSCs that promote angiogenesis tissue and a variety of differentiated strains, including immune cells [28].

The mechanism of the anti-fibrotic effect of the human AM can be explained by the inhibitory action of this tissue on TGF-β expression, which activates fibroblasts that, in turn, are responsible for the healing process. As a result, the fibrotic process is inhibited [12]. In this way, the AM, through the stimulation of tissue reconstruction

The internal imbalance of the body’s pro and antioxidants and known as oxidative stress. Moreover, it is important to note that in the process of wound healing, the oxidative stress and production of reactive oxygen species (ROS) increase, and these are very relevant in tissue regeneration. The two most important enzymes produced in the body are superoxide dismutase (SOD) and glutathione reductase (GRs), which are present when there is an excessive oxidation reaction [29].

Our data showed that carbonyl activity increased in the AM+ BMSCs group being almost 80% higher than the average of the other treatments, and was significantly decreased in the control group demonstrating that due to its antioxidant action. We also observed a significant increase in glutathione levels during treatment with AM group alone and the AM + BMSCs group and a decrease in the control group (*p* = 0.036).

However, evidence was found of higher levels in the AM+ BMSCs group and the AM group of GRs enzymes and SOD that helps the body cope with oxidative stress. Moreover, there was also a significant decrease in the activity of the antioxidant enzymes SOD and GRs, which are important antioxidant enzymes that can scavenge free radicals and protect the cell membrane. Studies show that there is a direct relationship between wound healing and the prevention of scar tissue formation depending on the decrease in oxidative stress [30,31].

These results are likely the consequence of the acute inflammation induced by flap wounds and the application of cells, in which the excessive production of ROS and inhibition of antioxidant enzymes occur. Abundant ROS production can lead to cellular damage and even death, while ROS production is very useful for the maintenance of homeostasis of the intracellular redox state [32,33].

Studies suggested that BMSCs partially reduced oxidative tissue damage and mitigated interleukin-1β-induced cell apoptosis. However, diverse types of methods and cells have been proposed for replacing wound healing. However, it is an ideal material with the potential for skin replacement still needs to be discovered [34,35].

After decellularization, the amniotic membrane offers several growth factors and other proteins that maintain the orientation of collagen fibers that are preserved during the process of such glycoproteins, proteoglycans, collagen, hyaluronic acid, and fibronectin [36,37].

In this study, the application of AM + BMSCs together demonstrated a positive effect by increasing the rate of collagen fiber formation, that are important for the quality of wound healing. These findings also can be explained by increasing oxygen levels that promote collagen synthesis, cell proliferation, and neoangiogenesis [38].

Thus, the decellularization process also increases biocompatibility due to the absence of secretion of human epithelial cells from soluble factors that promote inflammatory responses to cells that contribute to tissue repair and that collagen fibrils remain regularly organized [39]. Our previous study demonstrated the effects of this acellular biologic matrix for skin reconstruction using the rat model, suggesting that AM can promote skin repair, accelerating epithelialization mainly through paracrine signaling and differentiation cells [14].

Flap skin was performed, and the defects were changed by the AM + BMSCs cells. Histological examination showed the replacement of type III collagen for type I in the treatment group reached maturity more rapidly when compared to the control at 30 days of treatment after surgery. In this way, combining this therapy is a new perspective for treating tissue injury and promoting tissue regeneration. Previous studies proposed the benefits of mesenchymal or stem cell transplantation in scaffolding in promoting wound healing [40].

Thus, the combination of this therapy is a new perspective for the treatment of tissue injury that allows tissue regeneration. The acellular membrane promotes several therapeutic advantages can be by increasing only the acellular process but also the paracrine effects and growth factors inherent to the membrane, as well as its epithelialization and tissue regeneration [41]. This study was carried out, particularly in pre-clinical models. It will take considerable dedication to implement this purpose in clinical practices. In this study, AM facilitated the growth of human epithelial cells, suggesting that together AM+ BMSCs are an appropriate material for the skin and have a potential therapeutic tool for the reconstruction of a functional tissue surface.

In conclusion, AM + BMSCs can promote tissue regeneration. This effect may be attributed to the promotion of new collagen ECM synthesis by cell proliferation, and neoangiogenesis demonstrated an important role in the paracrine effects of the acellular matrix combined with cells.

## Figures and Tables

**Figure 1 biomedicines-11-01003-f001:**
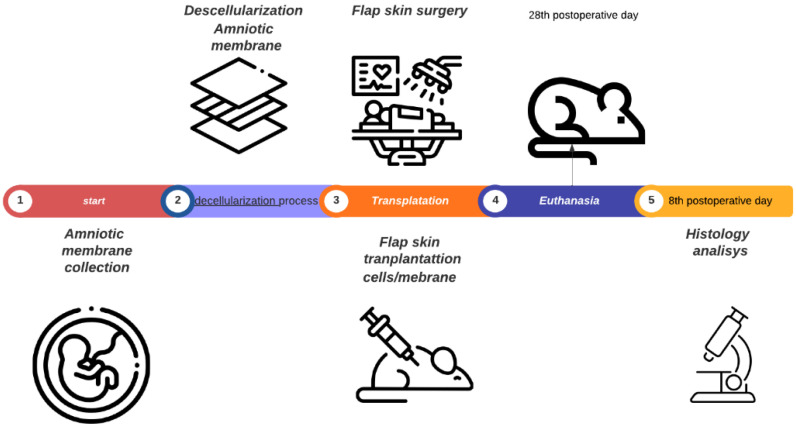
Experimental design: amniotic membrane collection, stem cell isolation, process membrane decellularization, flap skin surgery, implant, euthanasia, and histopathological analysis.

**Figure 2 biomedicines-11-01003-f002:**
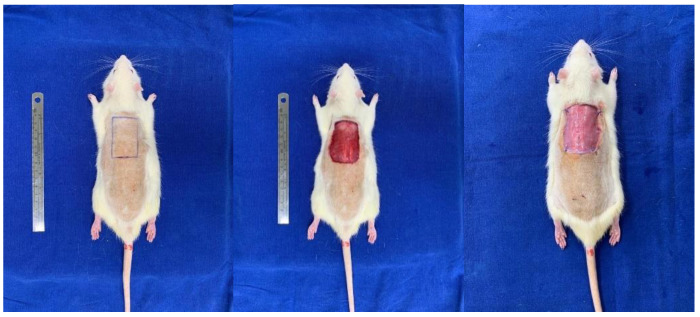
Flap skin exposition with a 30 × 50 mm (150 mm^2^) defect made with a scalpel.

**Figure 3 biomedicines-11-01003-f003:**
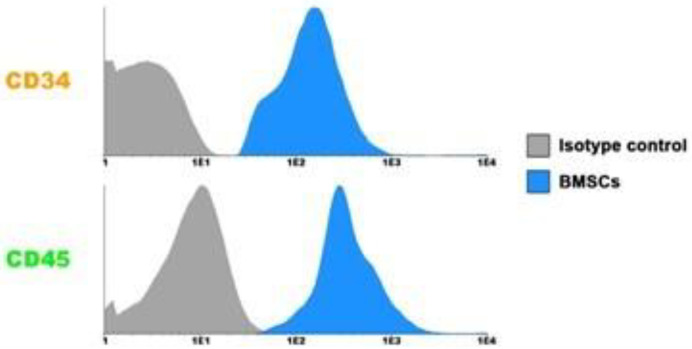
Characteristics of BMSCs. Flow cytometry Histograms of Bone marrow mononuclear cells. The histograms show that the population is positive for anti-CD45 and CD34.

**Figure 4 biomedicines-11-01003-f004:**
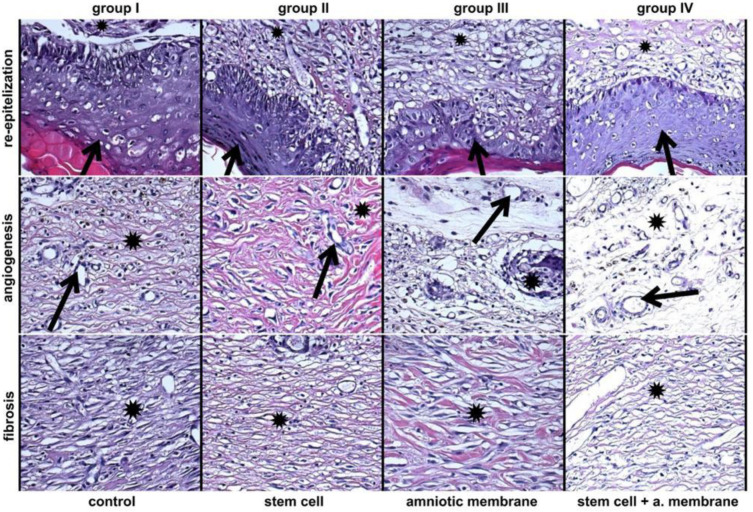
Depicts the signs of re-epithelialization, angiogenesis, and fibrosis with remodeling in four different groups: control (I), stem cell (II), amniotic membrane (III), and stem cell + amniotic membrane (IV). The samples were taken from the center of the lesion and analyzed using HE staining (hematoxylin-eosin) at 200× magnification. All groups, from I to IV, exhibit indications of complete re-epithelialization, as evidenced by a well-formed epidermis (indicated by a black arrow) over mature fibrous tissue (indicated by an asterisk). Angiogenesis is represented by numerous newly formed vessels (indicated by a black arrow) amidst fibrous stroma (indicated by an asterisk). Additionally, fibrosis with remodeling is represented by dense connective tissue containing thick, pink fibers that penetrate newly formed vessels (indicated by an asterisk).

**Figure 5 biomedicines-11-01003-f005:**
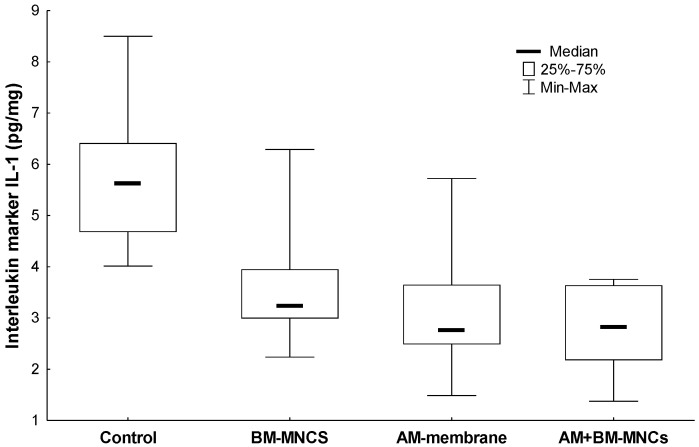
Effect of AM and BMSCs on Interleukin marker IL-1. Results are expressed as mean [interquartile range] ± SEM (N = 10). *p* <  0.05 denoted statistical significance in comparison to the control group. BMSCs: Bone Marrow Mesenchymal Stem Cells; *AM*: acellular human amniotic membrane.

**Figure 6 biomedicines-11-01003-f006:**
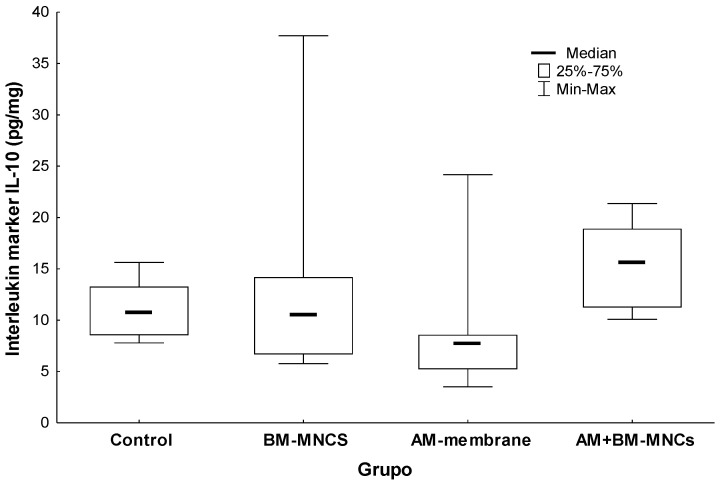
Effect of AM and BMSCs on Interleukin marker IL-10. Results are expressed as mean [interquartile range] ± SEM (N = 10). *p* < 0.05 denoted statistical significance in comparison to the control group. BMSCs: bone marrow mononuclear stem cells; *AM*: acellular human amniotic membrane.

**Figure 7 biomedicines-11-01003-f007:**
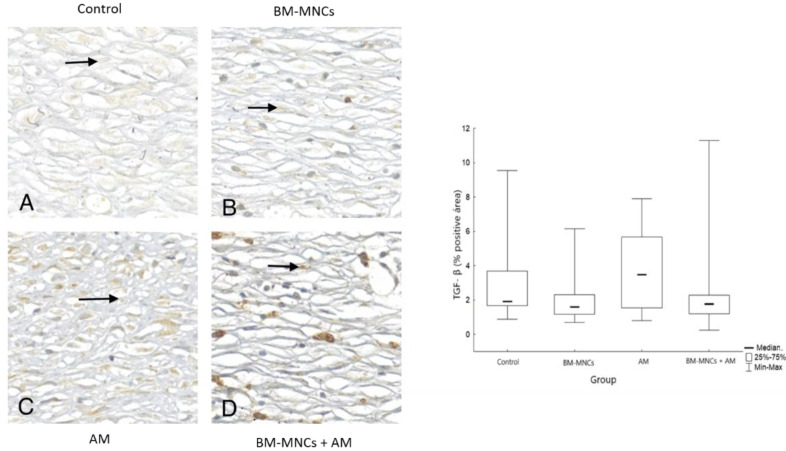
Representative immunohistochemical staining images of the expression of TGF-β showed the lowest percentages in the control (**A**), BMSCs (**B**), AM (**C**) and BMSCs + AM (**D**) groups on the 28th day of the experiments.

**Figure 8 biomedicines-11-01003-f008:**
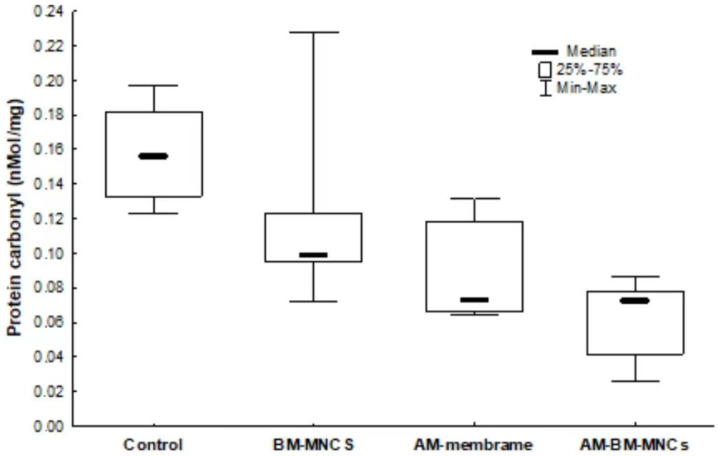
Effect of AM and BMSCs on carbonyl activity. Results are expressed as mean [interquartile range] ± SEM (N = 10). *p* < 0.05 denoted statistical significance in comparison to the control group. BMSCs: bone marrow mononuclear stem cells; *AM*: acellular human amniotic membrane.

**Figure 9 biomedicines-11-01003-f009:**
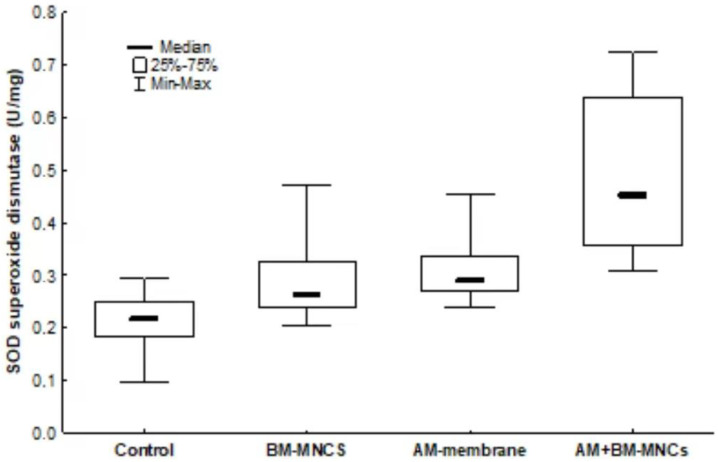
Effect of AM and BMSCs on SOD superoxide dismutase activity. Results are expressed as mean [interquartile range] ± SEM (N = 10). *p* < 0.05 denoted statistical significance in comparison to the control group. BMSCs: bone marrow mononuclear stem cells; *AM*: acellular human amniotic membrane.

**Figure 10 biomedicines-11-01003-f010:**
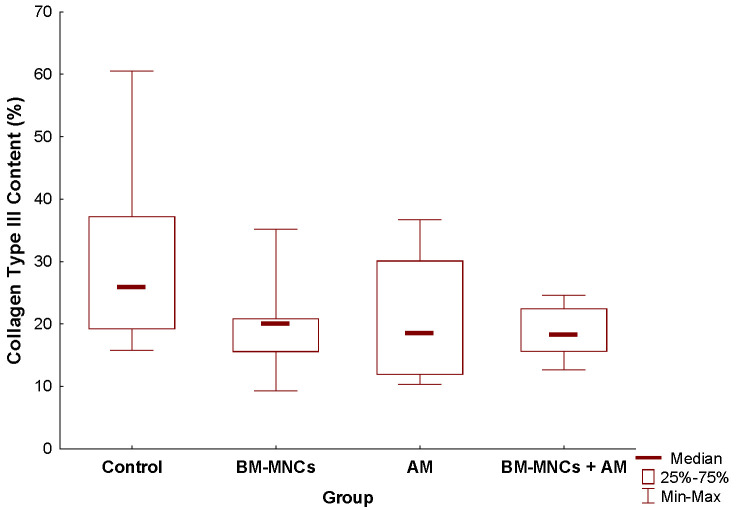
Collagen content analysis and flap skin area after 28 days. Collagen deposition quantified as Picrossirius Red positive areas, control group; Collagen type III content BMSCs group. Collagen type III content, acellular amniotic membrane. Collagen type III content, bone marrow mononuclear stem cells + acellular amniotic membrane group. Arrows indicate collagen type III. Results are shown as mean ± standard deviation. *p* < 0.0001. Images 20×, scale bar = 20 μm.

## Data Availability

Not applicable.

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
