# Peer review of "Acellular Biomaterials Associated with Autologous Bone Marrow-Derived Mononuclear Stem Cells Improve Wound Healing through Paracrine Effects"

_biomedicines, 2023, doi:10.3390/biomedicines11041003_

Round 1
Reviewer 1 Report (Previous Reviewer 3)
The article presents a subject with clinical importance: wound healing
1. Regarding oxidative damage, maybe it would be better to detect and other oxidative stress biomarkers.
Reviewer 2 Report (Previous Reviewer 2)
Dear Authors,
The resubmitted manuscript has been improved significantly. Good work!!
Reviewer 3 Report (Previous Reviewer 1)
All issues have been addressed well. I recommend the acceptance of this manuscript.
This manuscript is a resubmission of an earlier submission. The following is a list of the peer review reports and author responses from that submission.
Round 1
Reviewer 1 Report
In this article, researchers proved that acellular biomaterials associated with autologous bone marrow-derived mononuclear stem cells improve wound healing through paracrine effects. The data is informative and reliable. I recommend the acceptance of this article. However, before acceptance, some minor issues should be carefully addressed.
1. The authors should prove why it is the paracrine effects that matters for the stem cells.
2. The decellularization process of the amniotic membrane should be described in detail.
3. The HE staining of the healing tissue should be performed.
4. The authors should describe the mechanism of decellularized amniotic membrane.
Author Response
We thank the Referee for their interest in our work and for helpful comments that will greatly improve the manuscript and we have tried to do our best to respond to the points raised.
The Referee has brought up some good points and we appreciate the opportunity to clarify our research objectives and results. As indicated below, we have checked all the general and specific comments provided by the Referee and have made necessary changes accordingly to their indications.
Acellular biomaterials associated with autologous bone marrow-derived mononuclear stem cells improve wound healing through Paracrine Effects.
Isio Carvalho de Souza1, Aline Luri Takejima1, Rossana Baggio Simeoni1, Luize Kremer Gamba1, Victoria Stadler Tasca Ribeiro1, Katia Martins Foltz1*, Aloysio Enck Neto1, Meila Bastos de Almeida5, José Rocha Faria Neto1, Katherine A.T. Carvalho3; Paulo Cesar Lock da Silveira4, Ricardo Aurino Pinho2, Julio Cesar Francisco1 and Luiz Cesar. Guarita-Souza1. *
Reviewer 1
Thanks for commenting
- The authors should prove why it is the paracrine effects that matters for the stem cells.
Answer: Thanks for commenting will be corrected in the text.
- The decellularization process of the amniotic membrane should be described in detail.
Answer: Thanks for commenting will be corrected in the text. The decellularization process we use the protocol, which included mechanical agitation and the use of sodium dodecyl sulfate (SDS) protocol in a BioSAFE class II biological safety cabinet (Veco®). To prepare the acellular membrane, the amnion was washed thoroughly with PBS to remove blood and cellular debris. For this process, the membranes were removed from the solution (PBS) buffer phosphate pH 7.2 (Gibco) and treated with 0.01% SDS solution for 24 h at 37 °C with the aid of a mechanical shaker (Mesa Agitadora 109 M, Nova Ética Ltd., Lemesos, Cyprus). Then, the membranes were preserved in PBS at 4 °C, according to the methodology described in previous studies. In order to guarantee the decellularization of the acellular membrane, phase-contrast and fluorescence microscopic analyses of Hoechst staining were performed, confirming the absence of cells in the sample.
Blume GG, Machado-Junior PAB, Simeoni RB, Bertinato GP, Tonial MS, Nagashima S, Pinho RA, de Noronha L, Olandoski M, de Carvalho KAT, Francisco JC, Guarita-Souza LC. Bone-Marrow Stem Cells and Acellular Human Amniotic Membrane in a Rat Model of Heart Failure. Life (Basel). 2021 Sep 13;11(9):958. doi: 10.3390/life11090958. PMID: 34575107; PMCID: PMC8471644.
- The HE staining of the healing tissue should be performed.
Answer: Thanks for commenting will be corrected in the text. The section with hematoxylin and eosin (H&E will be included.
- The authors should describe the mechanism of decellularized amniotic membrane.
Answer: Thanks for commenting will be described in the text.
Reviewer 2 Report
Dear Authors,
The manuscript entitled Acellular biomaterials associated with autologous bone marrow-derived mononuclear stem cells improve wound healing through Paracrine Effects represents a good study in the field. However, major revisions are required before the manuscript will be processed to the next step of the publication process.
1) In the abstract section, . In the 26 full-thickness flap skin experiment of forty Wistar rats. Please include the number of Wistar rats in brackets, as you have performed previously in the abstract.
2) In the introduction section, please refer briefly the wound healing stages.
3) In the introduction section, Studies suggest that the wound healing process occurs through... Please add, the paracrine action of biomolecules such as. However, in the wound healing process besides the growth factors that you have mentioned also other biomolecules (e.g. cytokines, ECM proteins) are involved. Please refer to them briefly.
4) In the introduction section, Additionally, it has been demonstrated in clinical studies that the use of an acellular 68 amniotic membrane (AM). How an acellular AM can be produced. Please refer to the methods of production of acellular matrices (e.g. decellularization, crosslinking).
5) In the last paragraph, the authors refer to the BM-MNCs as stem cells. This is a misuderstanding , because the bone marrow contains stem cells, such as the Mesenchymal Stromal Cells. However, one method to obtain them is the isolation of MNCs and then substatially culture under in vitro conditions. So the authors must clearly distinguish between MNCs and MSCs. MSCs are multipotent stem cells, with specific characteristics, that must be discussed in this section here. Also please add as reference the following (https://doi.org/10.37349/ei.2021.00010)
6) Also the aim of the study is not clearly described here. Please revise it.
7) In the materials and methods section, 2.1 Animals, the authors should state that their experimental procedure fulfilled the declaration of Helsinki.
8) 2.3. Isolation and characterization of BMSCs / Acellular Amniotic Membrane Preparation. The authors have isolated stem cells from BM-MNCs, therefore the minimum criteria as has been described previously should be followed in order to well define the stem cells that are used in this study, Please follow the criteria as has been described by the ISCT here (10.1080/14653240600855905)
9) 2.6. Measurements of Cytokines, IL-1, IL10, and TGF-β levels
The levels of IL-1, IL-10 and TGF-β were quantified in tissue samples or blood samples?
10) In the results section, 3.1. Characterization of BM-MNC
The authors should show some figures of the isolated BM-MNCs. Did the authors performed cultures in order to establish their plastic adherent ability and fibroblastic morphology, as has been indicated by the ISCT. Also, flow cytometric diagrams showing the positive and negative expression of cd markers should be added. Differentiation assay (osteocytes, adipocytes and chondrocytes) of the isolated BM-MSCs should be performed in order to properly characterize these cells. The above experiments are strongly advised to be performed.
11) 3.2. Comparison of interleukin protein 1 (IL1) concentration in tissues adjacent to the skin 229 wound in rats, 3.3. Comparison of interleukin protein 10 (IL10) concentration in tissues adjacent to the skin 241 wound in rats, 3.4. Comparison of interleukin protein (TGF- β) concentration in tissues adjacent to the skin 252 wound in rats, 3.3. Oxidative damage markers. All these experimental procedures must be performed initially to animal's peripheral blood and then to tissue samples. Otherwise the results are not accurate enough.
12) 3.6. Collagen level analysis. The confirmation of collagen presence must be performed using either immunohistochemistry or immunofluoresence assay against collagen type I. Also, the quantification of collagen with picrosirius is not accurate enough. The quantification of collagen should be performed using the hydroxyproline assay kit (e.g. using the kit MAK008, from Sigma Aldrich).
13) Additionall, immunohistochemistry showing the presence of BM-MSCs on flap skin samples should be performed.
14) Additional, for the ECM remodelling also, quantification of sGAGs and elastin should be performed.
15) What about the immune reactions against the acellular AM. For this purpose immunohistochemistry against CD4, and CD11b should initially perform.
16) The discussion should be adjusted to the new results.
The authors should perform the majority of the comments, in order the quality of the manuscript to be improved and to proceed to the next step of the publication process.
Author Response
We thank the Referee for their interest in our work and for helpful comments that will greatly improve the manuscript and we have tried to do our best to respond to the points raised.
The Referee has brought up some good points and we appreciate the opportunity to clarify our research objectives and results. As indicated below, we have checked all the general and specific comments provided by the Referee and have made necessary changes accordingly to their indications.
Acellular biomaterials associated with autologous bone marrow-derived mononuclear stem cells improve wound healing through Paracrine Effects.
Isio Carvalho de Souza1, Aline Luri Takejima1, Rossana Baggio Simeoni1, Luize Kremer Gamba1, Victoria Stadler Tasca Ribeiro1, Katia Martins Foltz1*, Aloysio Enck Neto1, Meila Bastos de Almeida5, José Rocha Faria Neto1, Katherine A.T. Carvalho3; Paulo Cesar Lock da Silveira4, Ricardo Aurino Pinho2, Julio Cesar Francisco1 and Luiz Cesar. Guarita-Souza1. *
Reviewer 2
1) In the abstract section, . In the 26 full-thickness flap skin experiment of forty Wistar rats. Please include the number of Wistar rats in brackets, as you have performed previously in the abstract.
Answer: Thanks for commenting will be described in the text.
2) In the introduction section, please refer briefly the wound healing stages.
Answer: Thanks for commenting will be described in the text.
3) In the introduction section, Studies suggest that the wound healing process occurs through... Please add, the paracrine action of biomolecules such as. However, in the wound healing process besides the growth factors that you have mentioned also other biomolecules (e.g. cytokines, ECM proteins) are involved. Please refer to them briefly.
Answer: Thanks for commenting will be described in the text.
4) In the introduction section, Additionally, it has been demonstrated in clinical studies that the use of an acellular 68 amniotic membrane (AM). How an acellular AM can be produced. Please refer to the methods of production of acellular matrices (e.g. decellularization, crosslinking).
Answer: Thanks for commenting will be described in the text.
5) In the last paragraph, the authors refer to the BM-MNCs as stem cells. This is a misuderstanding , because the bone marrow contains stem cells, such as the Mesenchymal Stromal Cells. However, one method to obtain them is the isolation of MNCs and then substatially culture under in vitro conditions. So the authors must clearly distinguish between MNCs and MSCs. MSCs are multipotent stem cells, with specific characteristics, that must be discussed in this section here. Also please add as reference the following (https://doi.org/10.37349/ei.2021.00010)
Answer: Thanks for commenting your illustrative manuscript. The text will be correct. We consider that the acronyms to refer about all definition of the stem cells is well ample and percentage of Mesenchymal stromal cells (MSCs) represent 0.01–0.001% of total nucleated cells in bone marrow and are distinguished from “Mesenchymal Stem Cells”, which are characterized by restricted properties focused on self-renewal and differentiation potential. We separated the mononuclear cells fraction by Böyum methods.
Mallis P, Michalopoulos E, Chatzistamatiou T, Giokas CS. Interplay between mesenchymal stromal cells and immune system: clinical applications in immune-related diseases. Explor Immunol. 2021;1:112-39. https://doi.org/10.37349/ei.2021.00010
Böyum A. Isolation of mononuclear cells and granulocytes from human blood. Isolation of mononuclear cells by one centrifugation, and of granulocytes by combining centrifugation and sedimentation at 1 g. Scand J Clin Lab Invest Suppl. 1968;97:77–89.
6) Also the aim of the study is not clearly described here. Please revise it.
Answer: Thanks for commenting will be described in the text. But we to evaluate the paracrine effect were associate with acellular membrane.
7) In the materials and methods section, 2.1 Animals, the authors should state that their experimental procedure fulfilled the declaration of Helsinki.
Answer: Thanks for commenting will be described in the text.
8) 2.3. Isolation and characterization of BMSCs / Acellular Amniotic Membrane Preparation. The authors have isolated stem cells from BM-MNCs, therefore the minimum criteria as has been described previously should be followed in order to well define the stem cells that are used in this study, Please follow the criteria as has been described by the ISCT here (10.1080/14653240600855905)
Answer: Thanks for commenting will be described in the text.
Dominici M, Le Blanc K, Mueller I, Slaper-Cortenbach I, Marini F, Krause D, Deans R, Keating A, Prockop Dj, Horwitz E. Minimal criteria for defining multipotent mesenchymal stromal cells. The International Society for Cellular Therapy position statement. Cytotherapy. 2006;8(4):315-7. doi: 10.1080/14653240600855905. PMID: 16923606.
9) 2.6. Measurements of Cytokines, IL-1, IL10, and TGF-β levels
The levels of IL-1, IL-10 and TGF-β were quantified in tissue samples or blood samples?
Answer: Thanks for commenting will be described in the text. For quantified the cytokinies we used skin sample.
10) In the results section, 3.1. Characterization of BM-MNC
The authors should show some figures of the isolated BM-MNCs. Did the authors performed cultures in order to establish their plastic adherent ability and fibroblastic morphology, as has been indicated by the ISCT. Also, flow cytometric diagrams showing the positive and negative expression of cd markers should be added. Differentiation assay (osteocytes, adipocytes and chondrocytes) of the isolated BM-MSCs should be performed in order to properly characterize these cells. The above experiments are strongly advised to be performed.
Answer: Thanks for commenting will be described in the text. Will be presented in tex the diagrams and the figures of differentiation.
Representative histograms from the flow cytometry analysis showing surface marker expression on BMSCs for anti-CD45 and CD34.
11) 3.2. Comparison of interleukin protein 1 (IL1) concentration in tissues adjacent to the skin 229 wound in rats, 3.3. Comparison of interleukin protein 10 (IL10) concentration in tissues adjacent to the skin 241 wound in rats, 3.4. Comparison of interleukin protein (TGF- β) concentration in tissues adjacent to the skin 252 wound in rats, 3.3. Oxidative damage markers. All these experimental procedures must be performed initially to animal's peripheral blood and then to tissue samples. Otherwise, the results are not accurate enough.
Answer: Thank you for commenting, unfortunately we use only tissue, and we believe that our results corroborate the literature. In our next experiments, we will insert the analysis of the peripheral blood.
12) 3.6. Collagen level analysis. The confirmation of collagen presence must be performed using either immunohistochemistry or immunofluoresence assay against collagen type I. Also, the quantification of collagen with picrosirius is not accurate enough. The quantification of collagen should be performed using the hydroxyproline assay kit (e.g. using the kit MAK008, from Sigma Aldrich).
Answer: Thanks for commenting, the quantification the collagen by pricosirius this established by Experimental Pathology Laboratories, but we will insert into our next studies.
Blume GG, Machado-Junior PAB, Simeoni RB, Bertinato GP, Tonial MS, Nagashima S, Pinho RA, de Noronha L, Olandoski M, de Carvalho KAT, Francisco JC, Guarita-Souza LC. Bone-Marrow Stem Cells and Acellular Human Amniotic Membrane in a Rat Model of Heart Failure. Life (Basel). 2021 Sep 13;11(9):958. doi: 10.3390/life11090958. PMID: 34575107; PMCID: PMC8471644.
13) Additionall, immunohistochemistry showing the presence of BM-MSCs on flap skin samples should be performed.
Answer: Thanks for commenting, various studies have also demonstrated that BMSCs have the capacity to differentiate into certain cells in the neo-tissue. However, the quantification the other markers for collagen remodeling was assessed by MMP-8 expression, shown in another article from our group suggests that the transdifferentiation of BMSCs is involved in tissue repair.
Takejima AL, Francisco JC, Simeoni RB, de Noronha L, Garbers LAFM, Foltz KM, Junior PABM, Souza IC, Pinho RA, Carvalho KAT, Guarita-Souza LC. Role of mononuclear stem cells and decellularized amniotic membrane in the treatment of skin wounds in rats. Tissue Barriers. 2022 Apr 3;10(2):1982364. doi: 10.1080/21688370.2021.1982364. Epub 2021 Oct 6. PMID: 34612164; PMCID: PMC9067462.
14) Additional, for the ECM remodelling also, quantification of sGAGs and elastin should be performed.
Answer: Thanks for commenting, but we will insert into our next studies.
15) What about the immune reactions against the acellular AM. For this purpose immunohistochemistry against CD4, and CD11b should initially perform.
Answer: Thanks for commenting, but we will insert into our next studies.
16) The discussion should be adjusted to the new results.
Answer: Thanks for commenting the discussion will be reviewed with the proposed considerations.

Reviewer 3 Report
The article has an interesting topic, wound healing. but some changes must be performed
1. at line 79 appears the full name for BM-MNCs, so is not necessary to write once a gain the full name; e. g-line 84
2. the same situation is repeated for AM, appears full name, abbreviation,line 69, 85, 309
3. In the current study, immunohistochemical staining together with the cytokine level showed that the IL- 325 1 and IL-10 were overexpressed at the late stages of healing (28 days postoperative), these data corroborate with publications. This phrase confuses me: If IL-1 beta is a pro-inflammatory cytokine and IL-1 an anti-inflammatory one, how it is possible that these cytokines to be both overexpressed at the late stage of healing??????? Please explain
Author Response
We thank the Referee for their interest in our work and for helpful comments that will greatly improve the manuscript and we have tried to do our best to respond to the points raised.
The Referee has brought up some good points and we appreciate the opportunity to clarify our research objectives and results. As indicated below, we have checked all the general and specific comments provided by the Referee and have made necessary changes accordingly to their indications.
Acellular biomaterials associated with autologous bone marrow-derived mononuclear stem cells improve wound healing through Paracrine Effects.
Isio Carvalho de Souza1, Aline Luri Takejima1, Rossana Baggio Simeoni1, Luize Kremer Gamba1, Victoria Stadler Tasca Ribeiro1, Katia Martins Foltz1, Aloysio Enck Neto1, Meila Bastos de Almeida5, José Rocha Faria Neto1, Katherine A.T. Carvalho3; Paulo Cesar Lock da Silveira4, Ricardo Aurino Pinho2, Julio Cesar Francisco1 and Luiz Cesar. Guarita-Souza1.
Reviewer 3
- at line 79 appears the full name for BM-MNCs, so is not necessary to write once a gain the full name; e. g-line 84
Answer: Thanks for commenting will be corrected in the text.
- the same situation is repeated for AM, appears full name, abbreviation, line 69, 85, 309
Answer: Thanks for commenting will be corrected in the text
- In the current study, immunohistochemical staining together with the cytokine level showed that the IL- 325 1 and IL-10 were overexpressed at the late stages of healing (28 days postoperative), these data corroborate with publications. This phrase confuses me: If IL-1 beta is a pro-inflammatory cytokine and IL-1 an anti-inflammatory one, how it is possible that these cytokines to be both overexpressed at the late stage of healing??????? Please explain
Answer: Thanks for observation; we want to demonstrate that the IL-1 have an important role in the healing and that IL-1β was if present at the early phase of inflammation in the wound-healing process.
Round 2
Reviewer 2 Report
The authors have addressed the most of my comments succesfully. Well done!!